# Current and Future of Robotic Surgery in Thyroid Cancer Treatment

**DOI:** 10.3390/cancers16132470

**Published:** 2024-07-06

**Authors:** Joonseon Park, Kwangsoon Kim

**Affiliations:** Department of Surgery, College of Medicine, The Catholic University of Korea, Seoul 06591, Republic of Korea; 21700376@cmcnu.or.kr

**Keywords:** robotic thyroidectomy, transaxillary, transoral, BABA, conventional open thyroidectomy, comparison analysis

## Abstract

**Simple Summary:**

By synthesizing the latest research and clinical data, this review underscores the transformative potential of robotic surgery in thyroid cancer management and identifies key areas for further research to optimize patient outcomes. Our aim is to enhance the discourse on improving surgical care for thyroid cancer patients and to pave the way for future innovations in the field.

**Abstract:**

Thyroid cancer is among the most common endocrine malignancies, necessitating effective surgical interventions. Traditional open cervicotomy has long been the standard approach for thyroidectomy. However, the advent of robotic surgery has introduced new possibilities for minimally invasive procedures with benefits in terms of cosmetic outcomes, enhanced precision, comparable complication rates, and reduced recovery time. This study mainly reviewed the most widely used and well-known robotic thyroidectomy approaches: the transaxillary approach, the bilateral axillo–breast approach, and the transoral approach. This review examines the current status and future potential of robotic surgery in thyroid cancer treatment, comparing its efficacy, safety, and outcomes with those of conventional open cervicotomy. Challenges such as a longer operative time and higher costs exist. Future directions include technological advancements, tele-surgery, single-port surgery, and the integration of artificial intelligence. Robotic surgery holds promise in optimizing patient outcomes in thyroid cancer treatment.

## 1. Introduction

Thyroid cancer, although relatively rare, represents the most common malignancy of the endocrine system, with an increasing incidence over the past few decades [1,2]. This rise can be attributed to advancements in diagnostic techniques, such as high-resolution ultrasonography and fine-needle aspiration biopsies, as well as a heightened awareness among healthcare providers and patients [3,4,5]. Despite its generally favorable prognosis, thyroid cancer can present significant clinical challenges, particularly in terms of surgical management and the potential for recurrence [6].

Historically, open thyroidectomy (OT) has been the standard approach for the surgical treatment of thyroid cancer [7]. This procedure, while effective, involves a substantial neck incision, which can lead to considerable postoperative pain, visible scarring, and complications such as hypoparathyroidism and recurrent laryngeal nerve (RLN) injury [8,9,10,11]. These complications can significantly impact a patient’s quality of life (QOL), necessitating the exploration of alternative surgical techniques that can minimize these risks. In response to these challenges, minimally invasive surgical techniques have been developed and refined [12,13,14]. Among these, robotic-assisted surgery has emerged as a particularly promising approach. Robotic thyroidectomy (RT) utilizes advanced robotic systems to enhance the surgeon’s capabilities, providing superior dexterity and precision, and a three-dimensional visualization of the operative field [15,16,17]. These technological advancements have enabled surgeons to perform complex procedures through smaller, more cosmetically favorable incisions, potentially reducing postoperative pain and accelerating recovery [18,19].

The primary robotic approaches to thyroid surgery include the transaxillary (TA), transoral (TO), and bilateral axillo–breast approach (BABA) techniques [20,21,22,23,24,25]. The TA approach involves an incision in the armpit, avoiding a neck scar, while the TO approach involves incisions inside the mouth, further minimizing visible scarring. Both techniques have shown promising results in terms of cosmetic outcomes and patient satisfaction [9,24,25,26,27,28]. However, the adoption of robotic surgery in thyroid cancer treatment is not without its challenges. High costs [25,29,30,31] and the need for specialized training are significant barriers that must be addressed to fully realize the potential benefits of this technology [20,32,33,34,35,36,37,38,39].

This review provides a comprehensive overview of the current state and future prospects of robotic surgery in the treatment of thyroid cancer. We evaluate various robotic techniques in use, assessing their clinical outcomes, oncological efficacy, and potential advantages over traditional surgical methods. Additionally, we address the limitations and challenges of robotic thyroid surgery, including the cost, training requirements, and patient selection criteria. Furthermore, we explore future directions, highlighting emerging technologies and innovations poised to revolutionize the field, such as advancements in surgical robotics and the integration of artificial intelligence and machine learning to enhance surgical precision and decision making. This review aims to demonstrate how these advancements may significantly impact thyroid cancer surgery and improve patient outcomes.

## 2. Robotic Surgical Techniques in Thyroid Cancer Treatment

### 2.1. Transaxillary Approach

The TA approach to RT (TART) involves making an incision in the axilla (armpit) and creating a pathway to the thyroid gland. This approach avoids any visible scarring on the neck, which is a significant cosmetic advantage. The most common method is a single-incision, gasless TART using the robotic system, and the surgical procedure and steps are as follows [21,40,41]:(1)Position

Under general anesthesia in the supine position with endotracheal intubation, the neck is extended with a pillow under the shoulders. The arm on the side of the lesion is elevated and fixed to minimize the distance between the armpit and neck. This arm is then rotated nearly 180 degrees towards the head, placed on an armboard, and padded carefully.

(2)Skin flap creation

A 5–6 cm incision is made in the axilla, hidden when the arm is folded. A subplatysmal flap is created from the armpit to the anterior neck over the pectoralis major. Dissection proceeds between the two heads of the sternocleidomastoid (SCM) muscle, avoiding the internal jugular vein and common carotid artery. The omohyoid muscle is elevated, and the area between the strap muscle and thyroid gland is dissected until the upper pole and part of the contralateral thyroid gland are exposed. An external retractor is inserted through the axillary incision to maintain the working field.

(3)Robot docking

All four robotic arms are inserted via a single axillary incision. A 12 mm trocar for the camera is centrally placed, and an 8 mm trocar for the ProGrasp forceps is positioned to the right of the camera for a right-side approach and to the left for a left-side approach. The ProGrasp forceps should be parallel to the suction tube and close to the ceiling to avoid interference. The Maryland dissector and Harmonic curved shears are positioned on either lateral side.

(4)RT

The procedure begins by dividing the superior thyroid blood vessels. The ProGrasp forceps gently retract the superior thyroid pole caudally and away from the tracheoesophageal groove. The vessels are isolated with Maryland forceps and divided near the gland using a Harmonic scalpel. The dissection of the upper lobe continues until the superior parathyroid gland is identified and preserved. The lower lobe is then dissected, ensuring the RLN remains intact up to its innervation. Finally, the cervical trachea below the isthmus is identified, and the thyroid is separated from the trachea to complete the thyroidectomy.

The above steps are typically performed in the TART, with variations depending on the use of the da Vinci SP system [32,42,43] or CO_2_ insufflation [44,45].

(5)Single-port (SP) transaxillary robotic thyroidectomy

A smaller incision of 3 to 3.5 cm is used, facilitating the docking process as all four robotic arms are inserted through a single cannula, resulting in a shorter docking time. However, the flexible SP system does not yet support the Harmonic scalpel, requiring both arms to use the Maryland dissector with bipolar forceps or one arm to use a Monopolar blade for dissection [32,42,43].

(6)Gas-insufflation one-step single-port transaxillary (GOSTA) approach

In GOSTA, gas insufflation is maintained at 8 mmHg, and the skin flap is created using the robot. The SP endoscope is positioned at 12 o’clock, with arm 1 equipped with the Maryland bipolar forceps, arm 2 with the monopolar cautery, and arm 3 with the Cadiere forceps, regardless of the approach site [45].

### 2.2. Bilateral Axillo–Breast Approach

The BABA is another robotic technique that involves four small incisions: two in the axilla and two around the areola. This method offers a symmetrical and comprehensive approach to the thyroid gland, allowing for better access and visualization. The procedure generally involves the following [24,46]:(1)Position

The patient was positioned supine with a pillow under the shoulders to extend the neck, and both arms slightly abducted by the sides.

(2)Skin flap creation

Bilateral axillary and circumareolar ports (8–12 mm) were placed after the flaps were elevated using a tunneler. The flap is created with the superior border extending to the thyroid cartilage, the inferior border extending to 2–3 cm below the clavicle, and the lateral borders extending to the lateral margins of the SCM muscles on both sides. CO_2_ insufflation at 5–6 mmHg was used to maintain the working space. The midline was identified and divided with a hook until the thyroid gland was exposed.

(3)Robot docking

The camera is inserted through the right breast port, while the Harmonic scalpel is inserted through the left breast port. The grasper and retractors are inserted into the right and left axillary ports, respectively.

(4)RT

The isthmus is divided with the Harmonic scalpel to aid in lateral and posterior gland dissection and to provide the optimal visualization of the superior thyroid pedicle. Subsequently, the thyroidectomy was conducted with a clear visualization of the middle and inferior thyroid pedicles, the RLN, and the superior and inferior parathyroid glands. 

### 2.3. Transoral Approach

The TO approach to RT (TORT) is a relatively novel technique that involves accessing the thyroid gland through the oral cavity. This method entirely avoids external incisions, offering a scar-free neck. The procedure generally involves the following [23,47]:(1)Position

The patient is placed in a lithotomy position with the neck extended. The upper chest, axillae, and lower lip are exposed and painted with a povidone–iodine solution.

(2)Skin flap creation

A 2 cm inverted U-shaped incision is made at the midline of the lower lip after injecting 10 mL of epinephrine–saline solution (1:200,000). Hydrodissection is performed with 20–50 mL of diluted epinephrine solution into the oral vestibular area, chin, and anterior neck. Blunt tunneling widened the working space. A 12 mm trocar is inserted at the midline of the lower lip, and two 5 mm ports were placed near the lateral junctions of the lower lip. Insufflation pressure is maintained at 5–7 mmHg. Using a 10 mm videoscope, a working space is created down to the sternal notch and laterally to the SCM muscles’ borders.

(3)Robot docking

A 30-degree scope is inserted in the center, and the Maryland dissector and Harmonic scalpel are inserted through the lateral 5 mm ports. An additional 8 mm port is inserted in the right axillary area, and then the robot was docked.

(4)RT

The midline of the strap muscles is separated by a hook monopolar cautery. The isthmus is usually divided with a Harmonic shear, and lateral dissection is performed to separate the thyroid from the SCM. The superior pole and the cricothyroid muscle are dissected along an avascular plane, and the upper pole vessels are divided and sealed with a Harmonic shear. The upper parathyroid gland is identified and preserved. The thyroid gland is carefully dissected while retracting upward, and the RLN is identified at the insertion to the larynx near the Berry’s ligament and preserved. The thyroid is separated from the trachea after preserving the lower parathyroid gland. 

## 3. Advantages of Robotic Thyroidectomy

Table 1 compares the surgical and oncologic outcomes of conventional OT and RT. Below is the detailed interpretation of each category:

### 3.1. Cosmetic Benefits

Conventional OT requires a skin incision of at least 4–5 cm on the anterior neck, making visible scarring inevitable [7]. However, by using robotic surgery, the incision can be made in less visible areas such as the TA, BABA, or TO regions, resulting in higher satisfaction compared to conventional OT [24,25,26,27,28]. Lee et al. compared conventional OT with the TORT and TART and demonstrated that both the TORT and TART significantly improved cosmetic satisfaction scores in both the short and long term compared to OT (8.33 vs. 6.45; *p* < 0.0001, and 8.23 vs. 6.45; *p* < 0.0001, respectively) [9]. Kwak’s research reported that the rate of cosmetic satisfaction was significantly higher for the BABART compared to OT (84.9% vs. 59.3%; *p* < 0.001) [25]. 

### 3.2. Enhanced Precision 

Robotic surgery is particularly beneficial for navigating narrow and deep areas in thyroidectomy. Moreover, the robot allows the surgeon to perform operations from a seated position at a console, eliminating the need to manually handle surgical instruments directly. This minimizes tremors and reduces surgeon fatigue, enhancing the precision of instrument control [15,16,17]. The utilization of robotic technology offered enhanced visual clarity and three-dimensional viewing capabilities, improving surgical precision [16,63]. The parathyroid gland can be better preserved in RT compared to endoscopic thyroidectomy due to the more meticulous dissection with multi-articulated arms, which reduces interference from the camera [64].

### 3.3. Reduced Postoperative Pain

RT leads to significantly lower levels of postoperative pain compared to conventional OT, accompanied by the reduced utilization of analgesics [18,19,26]. In Ryu et al.’s comparison analysis, the TART resulted in lower Visual Analog Scale (VAS) scores compared to conventional OT. On postoperative days 1, 2, and 3, pain levels in the TART group were significantly lower than those in the OT group. (3.0 vs. 3.8; *p* = 0.001, 2.0 vs. 2.6; *p* = 0.005, and 1.3 vs. 1.7; *p* = 0.034, respectively) [19]. Fregoli et al. reported significantly lower VAS scores in the recovery room in the TART group compared to conventional OT (1.79 vs. 2.5; *p*  <  0.0001) [18]. However, the BABA and TORT often showed no significant difference in postoperative pain scores compared to OT. Cho et al. found no difference in pain scores between the BABA and OT [30]. In two studies by You et al. comparing the TORT and OT, VAS on postoperative day 1 showed no difference (2.64 vs. 2.19; *p* = 0.710) in one study [58], while the other study reported a higher VAS in the TORT (3.12 vs. 2.64; *p* = 0.043) [61].

### 3.4. Low Complication Rates

Common complications such as hypoparathyroidism, vocal cord palsy, and bleeding generally showed no significant difference between the RT and OT groups, as detailed in Table 1. However, some studies demonstrated differences in complication rates, with RT showing lower rates. In studies by Kim et al. and Paek et al. on the BABA approach, the incidence of transient hypoparathyroidism was lower in the BABA group compared to the OT group (23.0% vs. 36.3%; *p* = 0.01, and 2.8% vs. 10.5%; *p* = 0.042, respectively) [10,11]. Zhang et al.’s study showed that the incidence of both transient and permanent hypoparathyroidism was lower in the BABA group (17.9% vs. 29.7%; *p* = 0.016, and 2.0% vs. 6.8%; *p* = 0.043, respectively) [28]. In many studies, postoperative bleeding was observed to be 0% in robotic cases, although this finding was not statistically significant [25,26,27,29,52,54,56,57,60,62].

### 3.5. Postoperative Recovery

In most studies, no significant difference in hospital stay duration was observed between the RT and OT groups following surgery [10,11,19,25,26,27,29,30,48]. However, several QOL metrics that promote faster recovery were found to be superior with RT. In Lee et al.’s study, moderate and severe sensory changes, such as hyperesthesia or paresthesia in the neck, were significantly more frequent in the OT group compared to the RT group (62.1% vs. 19.4%, and 19.7% vs. 1.6%; *p* < 0.001). Postoperative swallowing discomfort was more frequent in the OT group compared to the RT group (7.9% vs. 4.1%; *p* < 0.001) [26]. Song et al. noted that the TORT offers superior postoperative voice outcomes when compared with OT. The TORT demonstrated a higher highest frequency (424.1 Hz vs. 299.9 Hz; *p* < 0.001) and a broader frequency range (281.5 vs. 167.8 Hz; *p* < 0.001) [60].

### 3.6. Oncologic Outcome

The oncologic outcomes and safety of RT have been proven to be comparable to those of OT. There is no significant difference in the number of harvested lymph nodes between the two methods [11,19,26,27,28,51,52,53,54,55,56,57,59,62], and postoperative thyroglobulin (Tg) levels also showed no difference [11,25,26,28,30,49,51,52,54,55,56,57,58,59]. Lee et al. demonstrated that there was no difference in recurrence rates and disease-free survival between the TART and OT in their postoperative comparisons (7.1% vs. 11.3%; *p* = 0.355, 9.29% vs. 88.7%; *p* = 0.938, respectively) [53]. 

### 3.7. Comparable Training Requirements

Surgeons require training to become proficient in robotic techniques. RT reduced surgeon fatigue and had a shorter learning curve compared to endoscopic thyroidectomy. The learning curve of endoscopic thyroidectomy was reported to be 30–90 cases for the TA approach [65,66,67], and 30–60 cases for the BABA [68,69], and 11–60 cases for the TO approach [70,71,72,73,74,75,76]. The learning curve of RT was reported to be 20–50 cases for the TA approach [20,32,33,34], 30–50 cases for the BABA [35,36,37], and 25–55 cases for the TO approach [38,39]. Several studies compared the learning curve of RT and endoscopic thyroidectomy. Lee et al. reported that the learning curve for the TORT was 25 cases, whereas, for the transoral endoscopic thyroidectomy vestibular approach (TOETVA), it was 71 cases [77]. Kiong’s comparison analysis between the TART and TA endoscopic thyroidectomy showed the TART has a shorter learning curve [78]. Lee et al. reported that the learning curve for the TART was 50 cases for total thyroidectomy and 40 cases for subtotal thyroidectomy [34]. Although most robotic surgeons are already well-trained endoscopic surgeons, robotic techniques do not require significantly more training and effort compared to traditional methods.

## 4. Limitations of Robotic Thyroidectomy

### 4.1. Longer Operative Time

In most previous studies, the operative time for RT was significantly longer compared to that of OT [19,25,26,27,28,29,48,49,50,51,52,54,60,61,62,79]. However, in a recent study examining the comparative effectiveness of RT versus OT in pediatric patients, no statistically significant disparity in operative time was observed (171.2 min vs. 182.6 min; *p* = 0.496) [53].

### 4.2. High Costs

RT requires significantly higher costs compared to OT [25,29,30,31]. In a cost analysis conducted by Broome et al. at a single institution, the expenses for OT were calculated at $2668, whereas RT incurred significantly higher costs at $5795 [31]. Cabot et al. reported that the cost of the TART was significantly higher than that of OT ($13,672 vs. $9028; *p* < 0.001) [29]. In studies comparing the BABA and OT, Kwak et al. and Cho et al. revealed that the BABA had significantly higher costs compared to OT ($9198 vs. $1489; *p* < 0.001, and $7632 vs. $2995; *p* < 0.001, respectively) [25,30]. In studies comparing the TORT and TOETVA, the cost of the TORT was significantly higher than that of TOETVA [80,81]. 

### 4.3. New Complications

While the incidence remains low, several complications have been reported with the TART, including brachial plexus injury, tumor seeding along the surgical track, seromas in the chest wall, and perforation or thermal damage to the axillary skin flap [82,83,84,85]. The TORT has been associated with unique complications such as oral commissure tearing, CO_2_ embolism, emphysema, mental nerve injury, skin perforation, burns, and bruising, which are uncommon, yet critical [86,87].

## 5. Comparisons between Different Approaches of RT

RT offers different advantages depending on the chosen approach, each with its unique benefits and limitations. First, the advantage of the TART is that it involves a single incision on the side with the tumor. However, it provides a limited view from one side only, which can make it challenging to identify the contralateral parathyroid glands and RLN during a total thyroidectomy [88]. Second, the BABA provides symmetrical views of each thyroid lobe, along with an excellent visualization of critical structures such as the parathyroid glands, RLN, and superior and inferior thyroid vessels. This approach allows the surgeon to operate on both thyroid lobes using consistent maneuvers [24]. Third, the TO approach has the significant advantage of leaving no scars, making it particularly suitable for patients prone to keloid or hypertrophic scarring. Studies have shown that patients undergoing the TORT report higher satisfaction in terms of cosmetic outcomes compared to those undergoing the BABA [89].

In Table 2, the surgical outcomes of RT are compared according to different approaches. The following are the results of a comparative analysis of surgical outcomes based on different approaches.

### 5.1. Postoperative Pain

Most of the comparisons are between the BABA and TORT [22,89,90,91,92]. Kim et al., Yang et al., and Chai et al. reported that pain levels were significantly higher with the the BABA (*p* = 0.021, *p* = 0.01, and *p* = 0.013, respectively) [89,90,92]. In contrast, Chae et al. and He et al. found that pain levels were significantly higher with the TO approach (*p*= 0.0046, *p* = 0.000) [22,91].

### 5.2. Operating Time

In four comparative studies on the BABA and the TO approach, two studies reported longer operating times for the BABA [90,92], while the other two found that the TO approach had longer operating times [22,91]. In Chae et al.’s comparative study, the TORT group had a longer operative time than the BABA group by 78.04 min (*p* < 0.001) [22]. Conversely, Yang’s study found that the operative time for the BABA was approximately 39 min longer (*p* < 0.01) [90].

### 5.3. Cosmetic Satisfaction

In specific, the TORT provided the greatest cosmetic advantage among the three methods due to the completely hidden scar. In a comparison study of the TORT and BABA, Kim et al. revealed that TORT patients had higher satisfaction levels regarding cosmetic outcomes compared to BABA patients [89].

## 6. Future Directions and Innovations in Robotic Thyroid Surgery

### 6.1. Technological Advancements

Future robotic systems are expected to offer an even greater precision, flexibility, and ease to use. Innovations such as smaller and more dexterous robotic arms, improved haptic feedback, and more intuitive control interfaces will enhance the surgeon’s ability to perform complex procedures with minimal invasiveness. The integration of advanced imaging technologies, such as high-resolution 3D imaging and augmented reality, will provide surgeons with a better visualization of the thyroid gland and the surrounding structures. This will aid in the more accurate identification of anatomical landmarks and potentially reduce the risk of complications. 

### 6.2. Tele-Surgery and Remote Assistance

Tele-surgery and remote assistance further extend the capabilities of robotic surgery by allowing expert surgeons to perform or assist in operations from remote locations. Utilizing high-speed internet and advanced communication technologies, tele-surgery enables real-time collaboration and intervention, expanding access to specialized surgical expertise globally. This technological integration holds promise for reducing surgical risks, improving access to care, and fostering global collaboration in the medical field.

### 6.3. Single-Port Surgery

The latest SP robotic system features a single cannula with three articulating instruments and a fully articulated 3D camera. This allows for smaller incisions and operations in tighter anatomical spaces [32,42,43]. Surgeons who are well-trained with the SP system are successfully completing challenging cases such as modified radical neck dissection, Graves’ disease, and multinodular goiter with minimal incisions. This approach not only ensures surgical and oncologic safety for patients but also enhances their quality of life by offering superior cosmetic outcomes.

Continued advancements in robotic technology, such as smaller and more versatile robotic instruments, may further improve the efficacy and safety of robotic thyroidectomy. The integration with augmented reality and artificial intelligence could enhance surgical planning and execution.

### 6.4. The Role of Fluorescence

Indocyanine green (ICG) during robotic thyroidectomy or parathyroidectomy for detecting and localizing parathyroid glands and central lymph nodes has been widely utilized [93,94,95,96]. In OT, near-infrared autofluorescence (NIRAF) imaging, which emits fluorescence without the need for ICG injection, has proven useful in thyroid surgery [97,98]. While these techniques have been applied sparingly in robotic surgery, they have been used to verify incidentally resected parathyroid glands post-surgery or to ensure the adequate removal of parathyroid adenomas. Incorporating NIRAF imaging devices into 3D camera equipment in the future could enhance the preservation or excision of parathyroid glands more effectively.

### 6.5. Potential Applications of Artificial Intelligence (AI)

Surgical planning and navigation coupled with real-time decision support hold significant promise in advancing the field of RT. Through the integration of AI, these technologies will revolutionize the preoperative planning process by precisely mapping out the patient’s thyroid anatomy and adjacent structures like parathyroid glands, the external branch of the superior laryngeal nerve, and RLN. AI algorithms analyze preoperative imaging data to identify optimal surgical pathways and potential challenges, allowing surgeons to tailor their approach to each patient’s unique anatomy. During the procedure, real-time decision support systems provide surgeons with immediate feedback and guidance, enhancing surgical precision and safety. Previous studies have proposed an AI for parathyroid glands and RLN recognition during open or endoscopic thyroid surgery [99,100], and research efforts are underway for its integration into RT. These systems can alert surgeons to deviations from the planned pathway, offer suggestions for optimal instrument positioning, and provide the real-time visualization of critical structures. By harnessing the power of AI in surgical planning and navigation, RT can achieve greater accuracy and efficiency, and, ultimately, improved patient outcomes. Moreover, AI-driven simulation platforms can provide advanced training for surgeons, offering realistic scenarios and feedback to help them improve their skills in RT. Virtual reality and AI can create immersive training environments that enhance learning and proficiency. Moreover, AI would be utilized to predict and prevent collisions between current robotic arms, optimize robot movements, and minimize tissue damage. Additionally, AI can assist in post-operative tracking and outcome prediction, continually aiding in making the best choices for surgical plans as they unfold.

### 6.6. Integration with Precision Medicine Approaches 

Depending on individual preferences and clinical situation, different methods can be chosen. If there is a history of head and neck surgery, radiation exposure, or the presence of oral inflammation or abscess, these conditions would constitute contraindications for the TO approach [101]. For patients with conditions like frozen shoulder that limit arm mobility or those with occupations requiring extensive arm use, alternative surgical approaches might be more suitable than the TA approach. In terms of cosmetic benefits, for someone who prioritizes the appearance of the breast, they may prefer the TART or TORT over the BABA. The TA approach has fewer incisions compared to the BABA and is superior as the scars are hidden in the armpit crease when the arm is down. However, if someone desires a scarless surgery that does not show any visible signs externally, then the TORT would be the appropriate choice. Since keloid scarring does not occur in the oral mucosa, the TO method is particularly beneficial for patients prone to keloid or hypertrophic scarring [89]. If a patient with lateral cervical lymph node metastasis requiring neck dissection expresses a preference for RT, the TA approach would be suitable, as it allows for a well-maintained lateral view and has demonstrated safety over several years [102,103].

## 7. Conclusions

In conclusion, RT represents a significant advancement in the management of thyroid cancer, offering improved cosmetic outcomes, functional benefits, and comparable oncological efficacy compared to traditional methods. Addressing the challenges of cost, training, and patient selection will be crucial for its broader adoption. With continued technological innovation and integration with precision medicine, RT is poised to play a pivotal role in the future of thyroid cancer treatment, ultimately enhancing patient care and outcomes.

## Figures and Tables

**Table 1 cancers-16-02470-t001:** Comparison analyses of outcomes between RT and conventional OT.

Author/Year	Sample Size (RT:OT)	Robotic Approach	Tumor Size (cm)	Operative Time (min)	Hospital Stay (days)	Pain (VAS)	Cosmetic Satisfaction	Transient RLN Injury (%)	Permanent RLN Injury (%)	Transient hypoPTH (%)	Permanent hypoPTH (%)	Bleeding (%)	No. of LN Retrieved	Postoperative Tg (ng/mL)
Lee et al., 2010 [27]	41:43	TA	8.3:8.9 (NS)	128.6:98.0 (*p* = 0.001)	2.5:3.2 (NS)	12.2:11.6 † (NS)	58.5:11.6 (*p* < 0.001)	2.4:0.0 (NS)	0.0:0.0 (NS)	19.2:15.3 (NS)	0.0:0.0 (NS)	0.0:2.3 (NS)	4.4:4.3 (NS)	NR
Lee et al., 2012 [48]	192:266	TA	0.6:0.6 (NS)	133.5:85.8 (*p* < 0.001)	3.3:3.3 (NS)	NR	NR	NR	2.0:0.0 (NS)	44.4:40.0 (NS)	0:1.12 (NS)	1.0:0.0 (NS)	4.6:5.7 (*p* = 0.004)	NR
Foley et al., 2012 [49]	11:16	TA	22.1:16.3 (NS)	232:109.3 (*p* < 0.001)	NR	NR	NR	NR	NR	1.0:1.0 (NS)	NR	NR	NR	1.0:0.8 (NS)
Landry et al., 2012 [50]	25:25	TA	NR	121:68 (*p* < 0.001)	NR	NR	NR	0.0:0.0 (NS)	20:16 (NS)	NR	NR	12:4 (NS)	NR	NR
Cabot et al., 2012 [29]	30:30	TA	0.7:0.8 (NS)	165.7:121.1 (*p* < 0.001)	5.1:5.1 (NS)	NR	NR	0:3.3	NR	33.3:26.7 (NS)	NR	0:0 (NS)	NR	NR
Kang et al., 2012 [51]	56:109	TA	1.14:1.49 (*p* = 0.004)	277.4:218.2 (*p* < 0.001)	6.0:8.0 (*p* = 0.008)	NR	NR	3.6:2.8 (NS)	0:0 (NS)	48.2:45.9 (NS)	0:0 (NS)	1.8:1.8 (NS)	37.3:39.4 (NS)	0.6:0.5 (NS)
Lee et al., 2013 [26]	62:66	TA	13.9:16.7 (NS)	271.8:208.9 (*p* < 0.001)	6.9:7.9 (NS)	1.5:2.0 (NS)	74.2:33.3 (*p* < 0.001)	3.2:4.5 (NS)	0:0 (NS)	38.7:34.8 (NS)	0:0 (NS)	0.0:0.0 (NS)	38.0:37.9 (NS)	0.6:0.5 (NS)
Yi et al., 2013 [52]	98:423	TA	0.8:0.8 (NS)	175.8:99.2 (*p* < 0.001)	4.0:3.4 (*p* < 0.001)	NR	NR	53.1: 43.0 (*p* = 0.046)	NR	53.1:43.0 (*p* = 0.046)	3.1:0.7 (NS)	0.0:0.5 (NS)	6.5:7.0 (NS)	0.5:0.4 (NS)
Ryu et al., 2013 [19]	45:45	TA	0.96:1.18 (*p* = 0.041)	121.8:99.8 (*p* < 0.001)	3.1:3.2 (NS)	3.04:3.82 (*p* < 0.001)	NR	NR	NR	NR	NR	NR	5.7:7.0 (NS)	NR
Lee et al., 2021 [53]	99:62	TA	1.8:2.5 (*p* = 0.010)	171.2:182.6 (NS)	3.6:4.8 (*p* < 0.001)	NR	NR	1.0:0.0 (NS)	0.0:1.6 (NS)	15.2:29.0 (NS)	2.0:3.2 (NS)	NR	43.0:59.3 (NS)	NR
Lee et al., 2021 [9]	40:40	TA	1.0:1.2 (NS)	184.9:132.1 (*p* < 0.001)	NR	NR	NR	5.0:5.0 (NS)	0.0:0.0 (NS)	27.8:25.0 (NS)	0.0:0.0 (NS)	0.0:5.0 (NS)	NR	NR
	40:40	TO	1.0:1.2 (NS)	185.6:132.1 (*p* < 0.001)	NR	NR	NR	2.5:5.0 (NS)	0.0:0.0 (NS)	9.1:25.0 (NS)	0.0:0.0 (NS)	25.0:5.0 (NS)	NR	NR
Kim et al., 2011 [54]	69:138	BABA	0.6:0.7 (*p* = 0.03)	196:81 (*p* < 0.001)	3.1:2.8 (NS)	NR	NR	1.4:0.7 (NS)	0.0:0.0 (NS)	33.3:27.5 (NS)	1.4:2.9 (NS)	0.0:0.0 (NS)	4.7:4.8 (NS)	0.8:0.8 (NS)
Kwak et al., 2015 [25]	206:634	BABA	0.8:1.0 (NS)	239:115 (*p* < 0.001)	3.4:3.3 (NS)	NR	84.9: 59.3 (*p* < 0.001)	0.5:0.9 (NS)	NR	14.6:15.0 (*p* = 0.29)	0.5:0.3 (NS)	0.0:0.9 (NS)	5.8:8.4 (*p* = 0.001)	20.2:41.2 (NS)
Kim et al., 2015 [10]	300:300	BABA	0.6:0.9 (NS)	175:115 (*p* < 0.001)	3.9:3.5 (NS)	NR	NR	2.6:1.3 (*p* = 0.08)	0:0.7 (NS)	23.0:36.3 (*p* = 0.01)	1.3:1.3 (NS)	0.3:0.3 (NS)	6.7:8.9 (*p* < 0.001)	NR
Cho et al., 2016 [30]	109:109	BABA	0.7:0.7 (NS)	290:107 (*p* < 0.001)	3.5:3.4 (NS)	3.8:3.8 (NS)	NR	6.4:5.5 (NR)	0.9:0.9 (NR)	33.0:26.6 (NR)	1.8:1.8 (NR)	0.9:0.9 (NR)	3.5:5.2 (*p* = 0.002)	0.2:0.3 (NS)
Chai et al., 2017 [55]	21:65	BABA	2.8:2.8 (NS)	165.1:93.5 (*p* < 0.001)	3.2:3.4 (NS)	NR	NR	19.0:9.2 (NS)	0.0:1.5 (NS)	19.0:33.8 (NS)	4.8:1.5 (NS)	NR	6.4:6.1 (NS)	0.3:0.3 (NS)
Bae et al., 2019 [56]	123:246	BABA	0.8:0.8 (NS)	198.4:123.5 (*p* < 0.001)	4.1:4.1 (NS)	NR	NR	31.7:35.8 (NS)	0.8:3.3 (NS)	31.7:35.8 (NS)	1.6:2.8 (NS)	0:0.8 (NS)	7.5:8.2 (NS)	1.2:1.1 (NS)
Paek et al., 2020 [57]	28:84	BABA	1.0:1.3 (*p* = 0.020)	382.3:195.9 (*p* < 0.001)	4.5:4.1 (NS)	NR	NR	10.7:7.1 (NS)	3.6:2.4 (NS)	7.1:10.7 (NS)	0.0:1.2 (NS)	0:0	36.5:40.0 (NS)	1.7:3.4 (NS)
Paek et al., 2018 [11]	71:305	BABA	0.83:0.81 (NS)	NR	NR	NR	NR	4.2:7.2 (NS)	0.0:0.7 (NS)	2.8:10.5 (*p* = 0.042)	0.0:0.0 (NS)	NR	7.9:8.5 (NS)	0.3:0.4 (NS)
Zhang et al., 2021 [28]	194:217	BABA	(NS)	162.71:93.51 (*p* < 0.01)	NR	NR	91.2 vs. 21.6 (*p* < 0.01)	0.5:1.0 (NS)	0.0:0.5 (NS)	17.9:29.7 (*p* = 0.016)	2.0:6.8 (*p* = 0.043)	0.5:2.1 (NS)	9.5:9.3 (NS)	0.6:0.5 (NS)
You et al., 2019 [58]	100:105	TORT	0.9:0.9 (NS)	209.8:97.6 (*p* < 0.05)	3.1:2.8 (NS)	2.6:2.2 (NS)	NR	1.0:0.0 (NS)	0.0:0.0 (NS))	22.2:19.5 (NS)	0.0:0.0 (NS)	1.0:0.0 (NS)	4.7:9.4 (*p* < 0.05)	0.3:0.3 (NS)
Tae et al., 2020 [59]	100:207	TO	1.0:1.5 (*p* = 0.001)	171.7:122.5 (*p* < 0.001)	NR	NR	1.5:2.9 (*p* < 0.001)	5.0:3.4 (NS)	0.0:1.0 (NS)	30.4:31.6 (NS)	4.3:1.3 (NS)	1.0:2.8 (NS)	6.3:6.6 (NS)	3.6:2.8 (NS)
Song et al., 2020 [60]		TO	1.3:1.5 (NS)	164.6:102.5 (*p* < 0.001)	NR	NR	NR	NR	NR	2.4:4.3 (NS)	0.0:2.1 ((NS)	0.0:2.1 (NS)	NR	NR
You et al., 2021 [61]	186:186	TO	0.8:0.6 (*p* < 0.05)	201.8:98.6 (*p* < 0.05)	2.9:2.6 (NS)	3.1:2.6 (*p* = 0.043)	NR	0.5:0.0 (NS)	0.0:0.0 (NS)	16.6:25.0 (NS)	0.0:0.0 (NS)	0.5:0.0 (NS)	5.0:8.6 (*p* < 0.05)	NR
Lee et al., 2023 [62]	100:100	TO	0.9:0.9 (NS)	196.1:109.3 (*p* < 0.001)	NR	NR	NR	0:4.0.0 (NS)	1.0:1.0 (NS)	10.0:8.0 (NS)	2.0:2.0 (NS)	0.0:3.0 (NS)	4.6:5.5 (NS)	NR

†: percentage of moderate pain on POD#1; TA, transaxillary; BABA, bilateral axillo–breast approach; RT, robotic thyroidectomy; OT, open thyroidectomy; NR, not reported; NS, not significant; LN, lymph node; RLN, recurrent laryngeal nerve; PTH, parathyroid hormone; Tg, thyroglobulin.

**Table 2 cancers-16-02470-t002:** Comparisons of outcomes between different approaches of RT.

Author/Year	Sample Size	Robotic Approach	Tumor Size (cm)	Operative Time (min)	Hospital Stay (days)	Pain	Cosmetic Satisfaction	Transient RLN Injury (%)	Permanent RLN Injury (%)	Transient hypoPTH (%)	Permanent hypoPTH (%)	Bleeding (%)	No. of LNs Retrieved
Kim et al., 2018 [89]	43:47	BABA:TO	0.9:0.7 (NS)	151.5:166.3 (NS)	3.5:3.8 (*p* = 0.020)	2.4:1.9 (*p* = 0.021)	3.4:3.7 (*p* = 0.044)	2.9:0.0 (NS)	0.0:0.0 (NS)	23.7:39.1 (NS)	43.5:46.1(NS)	0.0:0.0 (NS)	6.6:5.9 (NS)
Chae et al., 2020 [22]	56:14	BABA:TO	0.8:0.8 (NS)	221.6:299.16 (*p* < 0.001)	3.2:3.6 (NS)	2.3:2.7 (*p* = 0.005)	NR	10.7:0.0 (NS)	0.0:0.0 (NS)	0.0:0.0 (NS)	0.0:0.0 (NS)	0.0:0.0 (NS)	3.3:2.9 (NS)
Yang et al., 2020 [90]	316:248	BABA:TO	0.9:1.0 (NS)	243.8:204.1 (*p* < 0.01)	3.4:2.8 (*p* = 0.012)	3.3:2.7 (*p* = 0.01)	NR	2.5:1.2 (*p* = 0.01)	0.0:0.0 (NS)	16:7.1 (*p* = 0.03)	0.5:0.0 (NR)	0.0:0.4 (NS)	4.2:4.9 (*p* = 0.01)
He et al., 2022 [91]	50:49	BABA:TO	5.6:3.5 (*p* =0.037)	121.0:190.0 (*p* < 0.001)	10.1:9.5 (NS)	8.1:8.8 (*p* = 0.000)	NR	NR	NR	NR	NR	NR	8.3:7.9 (NS)
Chai et al., 2018 [92]	50:50	BABA:TO	1.1:1.0 (NS)	301.1:259.2 (*p* = 0.043)	3.9:3.4 (*p* = 0.011)	3.2:2.8 (*p* = 0.013)	NR	4.0:0.0 (NS)	0.0:0.0 (NS)	10.8:0.0 (NS)	0.0:0.0 (NS)	NR	4.0:4.5 (NS)
Lee et al., 2021 [9]	40:40	TO:TA	1.0:1.0 (NS)	185.6:184.9 (NS)	NR	NR	88.0:86.2 (NS)	2.5:5.0 (NS)	0.0:0.0 (NS)	9.1:27.8 (NS)	0.0:0.0 (NS)	2.5:0.0 (NS)	2.6:4.5 (NS)

TA, transaxillary; BABA, bilateral axillo–breast approach; RT, robotic thyroidectomy; NR, not reported; NS, not significant; LN, lymph node; RLN, recurrent laryngeal nerve; PTH, parathyroid hormone.

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
