# Peer review of "Current and Future of Robotic Surgery in Thyroid Cancer Treatment"

_cancers, 2024, doi:10.3390/cancers16132470_

Round 1

Reviewer 1 Report

Comments and Suggestions for Authors

Dear Authors,

Thank you for the opportunity to review this interesting papaer Robotic surgery is potential the best pratice for the surgeons and for the patients.

In thyroid surgery there are many exeprinces reported.

Your paper is  complete like review of results betewenn varius approachs and the open thyroidectomy.

I think the are necessary somes fixes and integrations.

In the abstrac you report that the robotic surgery  reduce the postoperative pain and recovery time as compared to  open surgery..  When you explain the table 1 there aren’t these advantages, so I think  It’s better to correct the abstract.

You don’t report the surgical limitations of robotic surgery. It’s important bias because in Occidental word, the indications for the robotic surgery is just about 10-20% of patients.

Above all, they don’t talk about the “new complications” that may show up with robotic surgery.

Author Response

In the abstrac you report that the robotic surgery  reduce the postoperative pain and recovery time as compared to  open surgery..  When you explain the table 1 there aren’t these advantages, so I think  It’s better to correct the abstract.

Response) Thank you for pointing this out. We agree with this comment. Therefore, we have corrected the abstract.
à However, the advent of robotic surgery has introduced new possibilities for minimally invasive procedures with benefits in terms of cosmetic outcomes, enhanced precision, comparable complication rates, reduced recovery time.

You don’t report the surgical limitations of robotic surgery. It’s important bias because in Occidental word, the indications for the robotic surgery is just about 10-20% of patients.

Above all, they don’t talk about the “new complictions” that may show up with robotic surgery.

Response) Following your comment, we have revised and added some . In section 4.3., we describe new complications associated with robotic thyroidectomy, including brachial plexus injury, tumor seeding along the surgical track, seromas in the chest wall, and perforation or thermal damage to the axillary skin flap. Section 5.6. discusses precision medicine approaches for appropriate indication selection, ensuring tailored patient management.

4.3. New complications

While the incidence remains low, several complications have been reported with TART, including brachial plexus injury, tumor seeding along the surgical track, seromas in the chest wall, and perforation or thermal damage to the axillary skin flap [82-85]. TORT has been associated with unique complications such as oral commissure tearing, CO2 embolism, emphysema, mental nerve injury, skin perforation, burns, and bruising, which are uncommon yet critical [86,87].

5.6. Integration with Precision Medicine Approaches

Depending on individual preferences and clinical situation, different methods can be chosen. If there is a history of head and neck surgery, radiation exposure, or the presence of oral inflammation or abscess, these conditions would constitute contraindications for the TO approach [101]. For patients with conditions like frozen shoulder that limit arm mobility or those with occupations requiring extensive arm use, alternative surgical ap-proaches might be more suitable than TA approach. In terms of cosmetic benefits, for someone who prioritizes the appearance of the breast, they may prefer TART or TORT over BABA. TA approach has fewer incisions compared to BABA and is superior as the scars are hidden in the armpit crease when the arm is down. However, if someone desires a scarless surgery that doesn't show any visible signs externally, then TORT would be the appropriate choice. Since keloid scarring does not occur in the oral mucosa, TO method is particularly beneficial for patients prone to keloid or hypertrophic scarring [89]. If a pa-tient with lateral cervical lymph node metastasis requiring neck dissection expresses a preference for RT, the TA approach would be suitable, as it allows for a well-maintained lateral view and has demonstrated safety over several years [102,103].

 We thank you and the reviewers for the insightful comments. We believe that our manuscript has been improved as a direct result of the review process. We hope that the revised manuscript is now suitable for publication in CANCERS.

Sincerely,

Joonseon Park, MD
Kwangsoon Kim, MD, PhD

Reviewer 2 Report

Comments and Suggestions for Authors

Nice job.

Would discuss an algorithm for technique selection between robotic types and what would be the patient selection for approach. 

Also would elaborate more on novel innovations to enhance robotics. The use of syncroseal in robotics. The role of fluorescence (ICG and other technologies)

The role of AI needs to be expanded. What specifically do the authors see down the pipeline that with help

The authors needs to also discuss the barriers to robotics and why adoption has not been more robust throughout the world. Its not just cost I would think.

The authors may also want to comment on radio frequency ablation and its evolving role in the algorithm.

Author Response

Comment1: Would discuss an algorithm for technique selection between robotic types and what would be the patient selection for approach. 

Response) Thank you for your feedback. Based on your comments, we have added the following content to section 5.5.

5.6. Integration with Precision Medicine Approaches

Depending on individual preferences and clinical situation, different methods can be chosen. If there is a history of head and neck surgery, radiation exposure, or the presence of oral inflammation or abscess, these conditions would constitute contraindications for the TO approach [101]. For patients with conditions like frozen shoulder that limit arm mobility or those with occupations requiring extensive arm use, alternative surgical ap-proaches might be more suitable than TA approach. In terms of cosmetic benefits, for someone who prioritizes the appearance of the breast, they may prefer TART or TORT over BABA. TA approach has fewer incisions compared to BABA and is superior as the scars are hidden in the armpit crease when the arm is down. However, if someone desires a scarless surgery that doesn't show any visible signs externally, then TORT would be the appropriate choice. Since keloid scarring does not occur in the oral mucosa, TO method is particularly beneficial for patients prone to keloid or hypertrophic scarring [89]. If a pa-tient with lateral cervical lymph node metastasis requiring neck dissection expresses a preference for RT, the TA approach would be suitable, as it allows for a well-maintained lateral view and has demonstrated safety over several years [102,103]

Comment 2: Also would elaborate more on novel innovations to enhance robotics. The use of syncroseal in robotics. The role of fluorescence (ICG and other technologies)

Response) Based on your advice, we have described "The Role of Fluorescence" in section 5.4. However, there have been no studies yet on the utilization of SynchroSeal among numerous energy devices in robotic thyroidectomy. Following your guidance, we will track and research the comparison of SynchroSeal and other energy devices in robotic surgery in the future. Thank you.

5.4. The Role of Fluorescence

Indocyanine green (ICG) during robotic thyroidectomy or parathyroidectomy for de-tecting and localizing parathyroid glands and central lymph nodes has been widely uti-lized [93-96]. In OT, near-infrared autofluorescence imaging (NIRAF), which emits fluo-rescence without the need for ICG injection, has proven useful in thyroid surgery [97,98]. While these techniques have been applied sparingly in robotic surgery, they have been used to verify incidentally resected parathyroid glands post-surgery or to ensure adequate removal of parathyroid adenomas. Incorporating NIRAF imaging devices into 3D camera equipment in the future could enhance the preservation or excision of parathyroid glands more effectively.

Comment 3: The role of AI needs to be expanded. What specifically do the authors see down the pipeline that with help

Response) Thank you for your comments. We have included the content in section 5.4 as discussed. Additionally, based on your feedback, we have added further content. The content is as follows:

5.5. Potential Applications of Artificial Intelligence (AI)

Surgical planning and navigation coupled with real-time decision support hold sig-nificant promise in advancing the field of RT. Through the integration of AI, these tech-nologies will revolutionize the preoperative planning process by precisely mapping out the patient's thyroid anatomy and adjacent structures like parathyroid glands, external branch of the superior laryngeal nerve and RLN. AI algorithms analyze preoperative im-aging data to identify optimal surgical pathways and potential challenges, allowing sur-geons to tailor their approach to each patient's unique anatomy. During the procedure, re-al-time decision support systems provide surgeons with immediate feedback and guid-ance, enhancing surgical precision and safety. Previous studies have proposed an AI for parathyroid glands and RLN recognition during open or endoscopic thyroid surgery [99,100], and research efforts are underway for its integration into RT. These systems can alert surgeons to deviations from the planned pathway, offer suggestions for optimal in-strument positioning, and provide real-time visualization of critical structures. By har-nessing the power of AI in surgical planning and navigation, RT can achieve greater ac-curacy, efficiency, and ultimately improved patient outcomes. Moreover, AI-driven simula-tion platforms can provide advanced training for surgeons, offering realistic scenarios and feedback to help them improve their skills in RT. Virtual reality and AI can create immer-sive training environments than enhance learning and proficiency. Also, AI would be uti-lized to predict and prevent collisions between current robotic arms, optimize robot movements, and minimize tissue damage. Additionally, AI can assist in post-operative tracking and outcome prediction, continually aiding in making the best choices for surgi-cal plans as they unfold.

Comment 4: The authors needs to also discuss the barriers to robotics and why adoption has not been more robust throughout the world. Its not just cost I would think.

Response) We agree. As per your advice, we have elaborated further on the barriers to robotic surgery beyond cost. In section 4.3, we have included new complications attributed to robotic thyroidectomy.

4.3. New complications

While the incidence remains low, several complications have been reported with TART, including brachial plexus injury, tumor seeding along the surgical track, seromas in the chest wall, and perforation or thermal damage to the axillary skin flap [82-85]. TORT has been associated with unique complications such as oral commissure tearing, CO2 embolism, emphysema, mental nerve injury, skin perforation, burns, and bruising, which are uncommon yet critical [86,87].

Comment 5: The authors may also want to comment on radio frequency ablation and its evolving role in the algorithm.
Response) As you commented, Radiofrequency ablation (RFA) is effective for treating thyroid tumors, providing a less invasive alternative to surgery without making an incision in the skin and directly targeting the thyroid. Therefore, this method seems unsuitable for robotic surgery because many RFAs are performed using hydrodissection without making an incision, allowing access to difficult-to-reach areas. However, combining RFA with robotic surgery could be advantageous when planning surgical re-operations to treat recurrent lymph nodes near the recurrent laryngeal nerve, where surgical removal is feasible. Moreover, so far, there have been no comparative studies between robotic surgery and RFA, hence a review of these procedures was not possible. We will reflect on future research and considerations regarding this topic in robotic surgery. Thank you for the feedback.

 We thank you and the reviewers for the insightful comments. We believe that our manuscript has been improved as a direct result of the review process. We hope that the revised manuscript is now suitable for publication in CANCERS.

Sincerely,

Joonseon Park, MD
Kwangsoon Kim, MD, PhD

Round 2

Reviewer 2 Report

Comments and Suggestions for Authors

Nice job